# Embryonated Chicken Tumor Xenografts Derived from Circulating Tumor Cells as a Relevant Model to Study Metastatic Dissemination: A Proof of Concept

**DOI:** 10.3390/cancers14174085

**Published:** 2022-08-23

**Authors:** Xavier Rousset, Denis Maillet, Emmanuel Grolleau, David Barthelemy, Sara Calattini, Marie Brevet, Julie Balandier, Margaux Raffin, Florence Geiguer, Jessica Garcia, Myriam Decaussin-Petrucci, Julien Peron, Nazim Benzerdjeb, Sébastien Couraud, Jean Viallet, Léa Payen

**Affiliations:** 1INOVOTION, 38700 La Tronche, France; 2University Claude Bernard Lyon, 69100 Villeurbanne, France; 3Department of Medical Oncology, Lyon Sud Hospital, Hospices Civils de Lyon, 69310 Pierre-Bénite, France; 4Centre de Recherche en Cancérologie de Lyon, INSERM 1052 CNRS UMR 5286, 69008 Lyon, France; 5Acute Respiratory Disease and Thoracic Oncology Department, Lyon Sud Hospital, Hospices Civils de Lyon, 69310 Pierre-Bénite, France; 6EMR-3738 Therapeutic Targeting in Oncology, Lyon Sud Medical Faculty, 69000 Lyon, France; 7Laboratoire de Biochimie et Biologie Moléculaire, Groupe Hospitalier Sud, Hospices Civils de Lyon, 69495 Pierre-Bénite, France; 8Circulating Cancer (CIRCAN) Program, Hospices Civils de Lyon, Cancer Institute, 69495 Pierre Bénite, France; 9Clinical Research Plateform, Institut de Cancérologie des Hospices Civils de Lyon, 69002 Lyon, France; 10Department of Pathology, Lyon Est Hospital, Hospices Civils de Lyon, 69677 Bron, France; 11Department of Pathology, Lyon Sud Hospital, Hospices Civils de Lyon, 69495 Pierre-Bénite, France; 12Laboratoire de Biométrie et Biologie Evolutive, Equipe Biostatistique-Santé, CNRS UMR 5558, Université Claude Bernard Lyon 1, 69100 Villeurbanne, France

**Keywords:** CTCs, CAM assay, metastasis, Alu sequences

## Abstract

**Simple Summary:**

Circulating Tumor Cells (CTCs) are heterogeneous and rare in the bloodstream, but responsible for cancer metastasis. Their in vitro or in vivo expansion remains a major challenge. The chicken Chorioallantoic Membrane (CAM) assay has proven to be a reliable alternative to the murine model, notably for tumor xenografts. We have developed a promising model of CTC-derived xenografts in the chicken CAM and demonstrated the feasibility of Next Generation Sequencing (NGS) analysis in this assay, with a genomic concordance between the in ovo tumor and the original patient’s tumor. We also evidenced metastatic dissemination from the xenograft in the chicken embryo’s distant organs. Further characterization of the in ovo tumors and metastases may provide new insights into the mechanisms of tumor dissemination. The development of a xenograft from a given patient’s CTCs, in a time frame compatible with managing the patient’s treatment, could also be a step forward towards personalized medicine.

**Abstract:**

Patient-Derived Xenografts (PDXs) in the Chorioallantoic Membrane (CAM) are a representative model for studying human tumors. Circulating Tumor Cells (CTCs) are involved in cancer dissemination and treatment resistance mechanisms. To facilitate research and deep analysis of these few cells, significant efforts were made to expand them. We evaluated here whether the isolation of fresh CTCs from patients with metastatic cancers could provide a reliable tumor model after a CAM xenograft. We enrolled 35 patients, with breast, prostate, or lung metastatic cancers. We performed microfluidic-based CTC enrichment. After 48–72 h of culture, the CTCs were engrafted onto the CAM of embryonated chicken eggs at day 9 of embryonic development (EDD9). The tumors were resected 9 days after engraftment and histopathological, immunochemical, and genomic analyses were performed. We obtained in ovo tumors for 61% of the patients. Dedifferentiated small tumors with spindle-shaped cells were observed. The epithelial-to-mesenchymal transition of CTCs could explain this phenotype. Beyond the feasibility of NGS in this model, we have highlighted a genomic concordance between the in ovo tumor and the original patient’s tumor for constitutional polymorphism and somatic alteration in one patient. Alu DNA sequences were detected in the chicken embryo’s distant organs, supporting the idea of dedifferentiated cells with aggressive behavior. To our knowledge, we performed the first chicken CAM CTC-derived xenografts with NGS analysis and evidence of CTC dissemination in the chicken embryo.

## 1. Introduction

Cancer remains a leading cause of death worldwide, especially in the lung, prostate, and breast [1]. The burden of its incidence and mortality is rapidly growing within the aging population and the increased multiple exposures to different cancer risk factors. It has led to intense scientific research, providing many advances in understanding molecular mechanisms. Recently, we have seen great improvements in the diagnosis and management of this heterogeneous disease [2]. Despite these recent advances, there is still a high attrition rate in drug development. Vastly improved in vivo models are therefore needed to drive the optimal treatment choice for patients, and to drive innovative drug discovery. Patient-derived tumor models in mice are amongst the most widely used. Despite their many strengths, these models are neither cost nor time effective for personalized medicine, therefore failing to predict the effectiveness of a treatment for a given patient in the medical timeframe [3]. 

Motivated by a global need for increased personalized medicine as well as compliance with the Reduce, Replace, and Refine (3Rs) policy, alternative models such as the chicken Chorioallantoic Membrane (CAM) xenograft are being developed. The CAM consists of the chorionic epithelium, a mesodermal layer, and the allantoic epithelium. This extra-embryonic membrane, connected to the embryo through a continuous extra-embryonic vessel system, is easily accessible for manipulation and observation [4]. It offers a rich environment in nutrients and embryonic growth factors, favorable for aggressive tumor development, with its highly vascularized membrane and natural immunodeficiency at engraftment (at EDD9). This in ovo model is well described to mimic intra-tumoral hypoxia [5]. In ovo xenografts have been performed for decades, from cell lines to patient tumor-derived xenografts, for several cancer types [6,7,8]. Comparison between CAM and patient tumors shows a good resemblance, for both histological and immunochemical analyses. The analysis of metastatic spreading, angiogenesis, tumorigenesis, and drug chemosensitivity testing is shown to be reliable [9,10,11]. For instance, in 1991, Shoin et al. developed 21 xenografts in the CAM from previously untreated malignant glioma. Xenografts and patients were treated with anti-cancer drugs. There was 78% agreement between the in ovo response and the corresponding patient response [12]. Its rapid readout could significantly improve drug development by shortening the gap between engraftment and evaluation of the treatment efficacy on the xenograft. 

Circulating cancer biomarkers including circulating tumor DNA (ctDNA) or Circulating Tumor Cells (CTCs) have emerged as a valuable resource for non-invasive diagnosis, putative prognosis evaluation, therapeutic monitoring, and for outcome prediction [13]. Importantly, 25% to 67% of patients with Non-Small Cell Lung Cancer (NSCLC) cannot benefit from a re-biopsy analysis during first progression. Even though the re-biopsy is possible, molecular interpretation of the results is limited by their spatial and temporal tumor heterogeneity [14]. CTC exploration provides a snapshot of tumor heterogeneity with a demonstrated concordance between tissues and CTC genomic profiles [15]. Single cell RNA sequencing (scRNAseq) recently demonstrated a deeper level of heterogeneity, with several different CTC subpopulations including epithelial, epithelial–mesenchymal, mesenchymal, and stem-like phenotypes [16]. Overall, this new perspective could allow identification of the subpopulations leading to metastatic spreading and treatment resistance, and thus cancer progression or recurrence [16,17]. The challenge is to further explore these tumor phenotypes and, what is more challenging, their reliable expansion. This will allow the emergence of new individualized tumor models, representative of the patient’s tumor heterogeneity.

Living CTCs are rare in the bloodstream. Enrichment of this small number of living CTCs to perform all these analyses remains a huge challenge. Several methods were used, mostly targeting physical size or density, or expression of tumor biomarkers or immune cell biomarkers [18]. The only FDA-certified technique for enumeration of CTCs in routine use is capturing dead CTCs with the CellSearch^®^ approach using anti-EpCAM-coated magnetic beads [16]. As this method only targets Epithelial Cell Adhesion Molecule (EpCAM)-positive CTCs, it is not able to recover the whole diversity of CTC subtypes [19,20,21]. Moreover, the Paraformaldehyde Fixation (PAF) step used in this approach is not compatible with later in vivo expansion of CTCs. Another method for isolation of CTCs, using a filtration technique, is Isolation by Size of Epithelial Tumor (ISET). Despite its high sensitivity, strong criteria for cytopathological analysis of CTCs and sensitivity for small cell detection are missing [22,23]. In addition, specific protocols to keep CTCs alive exist, but no application has been deeply described in the literature [24,25]. The RosetteSep™ device combines the use of antibodies to change the density of unwanted cells, followed by their removal using density gradient centrifugation. This method has shown good performance with different cancer types. Its feasibility for ex vivo culture in breast cancer has already been demonstrated [26]. Finally, microfluidic-based methods are among the most efficient. We selected the ClearCell FX1 (from Biolidics) instrument to allow a higher rate of living cell enrichment. Based on spiral microfluidics using inertial and Dean drag forces, it separates cells according to their size and plasticity [27]. This technique allows a high recovery rate with more than 80% sensitivity and provides intact and viable cells for culture and in vivo models [27]. 

Despite recent advances, in vitro and in vivo CTC expansions remain challenging, especially for long-term culture and xenografting [28,29]. A few studies reported the development of CTC-derived xenograft models in immunocompromised mice [28,30,31]. Expansion requires a high blood CTC concentration and the most appropriate enrichment method. Culture conditions for the generation of xenografts have not yet been refined [29]. Studies reporting in vitro or in vivo drug testing are rare and non-standardized, often failing to predict the treatment response, and are therefore not yet suitable for guiding clinical decisions [28,32]. The CAM model seems to be a very good choice with its rapid growth, low cost, reproducibility, and reliability in reproducing human tumors. As successfully described by Pizon et al., the in ovo CTC-Derived Xenograft (in ovo CDX) could provide an avatar of the patient’s tumor, allowing drug screening while respecting a short timeframe to allow timely guidance for therapeutic decisions [33]. 

Overall, we propose an exciting and ambitious project evaluating whether freshly isolated CTCs from a patient with a metastatic cancer could provide a reliable tumor model after a CAM xenograft. Our primary endpoint was the in ovo engraftment rate, defined as a tumor proven via a histopathological examination. The secondary endpoint was the evaluation of the metastatic dissemination of the CTCs in the chicken embryo, through genomic analysis. Potential associations between clinical features, initial tumor features, CTC count, and the engraftment rate were explored.

## 2. Materials and Methods

### 2.1. The Study Population 

Thirty-five patients (Figure 1) were recruited from the pulmonology and oncology departments at Lyon University Hospital (LHU), from 2018 to 2022: 6 in the prostate cohort, 6 in the breast cohort, and 23 in the lung cohort. Inclusion criteria were: age over 18 years old, with histologically proven lung, prostate (adenocarcinoma without endocrine contingent), or breast cancer, at a metastatic stage. Lung cancer patients must be negative for EGFR, ALK, and ROS-1 somatic alterations, with an indication for chemotherapy. We included only castration-resistant prostate cancers, with an indication for chemotherapy. Breast cancers included were refractory to previous endocrine therapies or were triple negative breast cancers, with chemotherapy indications. Written consent was obtained for all patients before enrollment. 

### 2.2. Ethics

SENCIRTEG is an ancillary study of the CIRCAN program, which is non-interventional, prospective, and uses a biological cohort designed for setting up various diagnostic tests on circulating free DNA and CTC in cancer patients at LHU. CIRCAN-ALL and its amended version, including SENCIRTEG, are classified as non-interventional studies by the Lyon Sud-Est IV ethics committee (L15-188, the 4 November 2015, amended on 16 September 2016, L16-60). SENCIRTEG falls within Reference Methodology N3 of the French National Commission for Data Protection and Freedom of Information (Commission Nationale de l’Informatique et des Libertés, CNIL) for which the Hospices Civils de Lyon (HCL) has signed a commitment for compliance and respecting the RGPD (HLC register MR003_17_103). All included patients were fully informed and signed a standardized consent form.

### 2.3. Sample Collection 

Peripheral blood tests were performed during scheduled hospital visits. Samples were collected as part of the CIRCAN-ALL program, a prospective translational LHU program for the evaluation of tumor biomarkers in liquid biopsies. For each included patient, 3 × 10 mL of whole blood were collected in K2EDTA tubes (BD, 367525, 18 mg), used for CTC counting, in ovo CTC expansion, and for cell-free DNA (cfDNA) analysis. The samples were sent to our laboratory department at Room Temperature (RT).

### 2.4. CtDNA Collection and Library Preparation for DNA Sequencing

Total blood samples were centrifuged for 10 min at 1600× *g* and the cell pellet was conserved for CTC enrichment. The supernatant was then centrifuged at 6000× *g* for 10 min, and the resulting plasma was stored at −80 °C until cfDNA extraction and molecular analysis. cfDNA was extracted using the QIAamp Circulating Nucleic Acid Kit (Qiagen, Cat No 55114, Valencia, CA, USA), with a Qiagen vacuum manifold following the manufacturers’ instructions. Circulating cfDNA samples were quantified using a Qubit™ 4 Fluorometer (Thermo Fisher Scientific, Waltham, MA, USA, Invitrogen™, Cat No Q33226) with the Qubit™ dsDNA HS Assay Kit (Thermofisher Scientific, Waltham, MA, USA, Cat No 32854). We performed the same method for DNA extraction from fresh CTCs of patient 13, following the enrichment step through the Clearcell FX device.

For Next Generation Sequencing (NGS) library preparation, 10–50 ng of DNA (cfDNA, DNA extracted from fresh CTCs or DNA extracted from the CAM tumors) were used, using a custom capture-based technology provided by Sophia Genetics and performed according to the manufacturer’s instructions. The custom panel covered 78 genes involved in cancer (such as TP53, KRAS, or ATM). The libraries were sequenced on NextSeq 550 (Illumina, San Diego, CA, USA) in 2 × 150 paired-end runs. The subsequent Variant Call Files were subjected to cross-sample background filtering, with potential artefacts removed below 3 standard deviations of the mean background noise for each position. Filter criteria for variant calling were set to an absolute number of mutated allele read counts ≥40, a minimal total depth >2000×, and a Mutant Allele Fraction (MAF) threshold ≥0.5% [34].

### 2.5. Enrichment of CTCs or Mimicking CTCs from Blood Samples

After collection of plasma for subsequent ctDNA molecular analysis using NGS, the cell pellet containing: (i) red blood cells, (ii) white blood cells, and (iii) CTC or spiked fluorescent mimicking CTCs (PC3 cells (CRL-1435™, ATCC), A549 cells (CCL-185™, ATCC)) was resuspended in a volume of Phosphate-Buffered Saline (PBS) equivalent to initial total plasma volume. Finally, RBCs were lysed according to the purchaser’s recommendation (Biolidics, Singapore), and the remaining pellet was processed on the ClearCell FX1 spiral microfluidic device. The output collection tube was previously filled with 3 mL of Dulbecco’s modified Eagle’s medium (without newborn calf serum). The CTC sample within the CTChip^®^ was run through the 30 min program. The cell enrichment was centrifuged at 500× *g* for 5 min, before tissue culture at 37 °C in a humidified 5% CO_2_ atmosphere, into an Ultra-Low Adherence (ULA) cell dish for 2–3 days within the previously mentioned medium, before engrafting in the CAM model. An independent enrichment was carried out for immunofluorescence staining. The detailed procedure for CTC enrichment using the ClearCell device was previously published. As described in this work, we also spiked the fluorescent mimicking CTCs cells with CellTracker™ Green CMFDA Dye (Thermo Fisher Scientific, Waltham, MA, USA, cat No C7025,) and a predefined number of cells (using fluorescence microscopy for absolute enumeration) was added into the normal blood of a healthy donor, and run through the ClearCell FX1, with the 30 min program [14].

### 2.6. CTC Enumeration and Characterization 

CTC enumeration and characterization were performed through immunostaining with the same procedure as detailed in a previous publication [14]. The slides were scanned using the Lionheart X/Y motorized fluorescent microscope at 20×. Three filters were used to detect the fluorescent signal: DAPI (excitation peak at 359 nm and an emission peak at 457 nm) for nucleus staining, Alexa488 (excitation peak at 499 nm and an emission peak at 520 nm) for CD45 (anti-CD45 antibody, 1:500 rat anti-human, MA5-17687, ThermoFisher), CD15 (anti-CD15 antibody, 1:250 mouse anti-human, 560172, BD Bioscience), and CD41 (anti-CD41 antibody, 1:40 mouse anti-human, 303723, BioLegend, San Diego, CA, USA), and CY5 (excitation peak at 651 nm and an emission peak at 670 nm) for PD-L1 (anti-PD-L1 antibody, 1:200 rabbit anti human, 86744S, Cell Signaling Technology). Parametric settings included exposure time, LED intensity, gain, and focus. The latter two were adjusted with the associated software (Gen5™ version 3.09, Biotek, Santa Clara, CA, USA). We considered here as CTCs the negative cells for CD45, CD15, and CD41 but positive for DAPI. Unfortunately, the immunostaining technique was under development at the beginning of the study, and thus not used. CTC counts and PD-L1 evaluation were therefore only available for 16 patients.

### 2.7. Conservation and Transfer of Samples 

CTCs were placed after enrichment in ULA culture microplates with Dulbecco’s modified Eagle’s medium (without newborn calf serum). We defined a 48–72 h delay between CTC enrichment and in ovo engraftment (culture time). 

### 2.8. In Ovo Xenograft

After 48–72 h of in vitro culture (non-adherent conditions), purified CTCs were washed once in 1 mL of Roswell Park Memorial Institute (RPMI) medium, and resuspended in a mix (volume:volume, 1:1) of complete medium (RPMI, 10% FBS and penicillin/streptomycin) and Matrigel (Corning, Matrigel^®^ Growth Factor Reduced (GFR) Basement Membrane Matrix, LDEV-free; #354230).

Fertilized White Leghorn eggs were obtained from Couvoir Hubert, France. Eggs were incubated at 37.5 °C with 50% relative humidity for nine days. On the 9th embryonic development day (EDD9), a small hole was drilled through the eggshell into the air sac, and a 1 cm^2^ window was cut into the eggshell above the CAM. 

CTCs were then grafted onto the CAM by depositing 50µL of the CTC suspension. Frequent evaluations of tumor growth and confirmation that the egg was still alive were performed, through the window in the eggshell.

The time between engraftment and tumor occurrence was recorded. At EDD18, the tumors were harvested, stored in 4% paraformaldehyde for at least 48 h, and embedded in paraffin for further histological analysis. 

In parallel, and following the same protocol, lung cancer cells from the HCC827 cell line (CRL-2868, ATCC, Manassas, VA, USA) were engrafted to obtain a CAM tumor xenograft used as a positive control for the CTC experiments. 

According to French legislation, no ethical approval is required for scientific experiments using oviparous embryos (Decree No. 2013-118, 1 February 2013; art. R-214-88). 

### 2.9. Tumor Analysis 

#### 2.9.1. Anatomopathological Analysis 

Pathological analyses were performed at the LHU’s department of anatomopathology. Two independent observers evaluated the slides and scored the presence of a visible macroscopic nodule and its size if present. For all the slides with a microscopic tumor, they scored the presence of cell proliferation, mitosis, nuclear polymorphism, tumor stroma, and necrosis. They also described the degree of differentiation of the tumor, the cohesive or non-cohesive nature of the cells, and their architecture, as well as the cell shape. The characteristics of the stroma were reported as inflammatory or fibrous, and the vascularization as poor or abundant. Based on these morphological criteria, they classified the slides as having the presence of tumor, probable presence of tumor, or absence of tumor. 

The histological sections were deparaffinized and hydrated in a series of xylene and graded alcohol, then stained with Hematoxylin and Eosin (H&E) according to a standard protocol.

Immunochemistry was performed on a sample of formalin-fixed paraffin-embedded tissue sections. Briefly, formalin-fixed paraffin-embedded 4μm thick sections were first deparaffinized in xylene and rehydrated in alcohol. The endogenous peroxidase activity was blocked (Ventana Medical Systems, Tucson, AR, USA) before antigen retrieval. The ULTRA Cell Conditioning Solution (ULTRA CC1) from Ventana Medical Systems was then used for antigen retrieval. Immunohistochemical staining was carried out on an automated immunostainer, with primary antibodies. This was followed by application of the avidin–biotin–peroxidase complex technique. Reactions were developed with diamino-3,3′-benzidine tetrahydrochloride substrate solution (SIGMAFAST; Sigma-Aldrich, Tucson, AZ, USA). The tissues were counterstained with hematoxylin. The primary antibodies and final dilutions were: anti-Thyroid Transcription Factor-1 (TTF-1) (clone 8G7G3/1 Ventana Medical Systems, prediluted by the manufacturer) for in ovo tumors obtained from lung cancer patients; and anti-Cytokeratin AE1/AE3 (CK3) (1:100; Glostrup, Denmark) for in ovo tumors obtained from lung cancer and breast cancer patients.

Alongside these CDX analyses, we used as a positive control the tumors obtained after the engraftment the HCC827 lung cancer cell line (CRL-2868, ATCC, Manassas, VA, USA) on the chicken embryo CAM.

#### 2.9.2. Quantitative PCR for Alu Sequence Detection 

To characterize tumor cell dissemination in chicken tissues, a qPCR analysis was carried out using specific sets of primers against human Alu sequences. The presence of many copies of Alu sequences in human genomic DNA increases qPCR sensitivity, allowing detection of a few human cells in a chicken tissue [11]. 

We chose Formalin-Fixed, Paraffin-Embedded (FFPE) CAM tumors of more than 1 mm in the largest diameter at histopathological analysis. An area with >70% tumor cells was annotated for macrodissection. Total nucleic acid was extracted using the manual extraction process with a QIAamp DNA Mini QIAcube Kit (Thermo Fisher Scientific, Waltham, MA, USA, Qiagen^®^, cat no 51326). The quantification of DNA was performed using the Qubit fluorometer (Life Technologies, Carlsbad, CA, USA). 

We performed quantitative PCR (qPCR) of Alu sequences to evaluate the human DNA concentration in the CAM tumors. We first diluted the samples with sterile H_2_O to obtain 50 ng DNA inputs for tumors generated from patient’s CTCs and for the negative control (chicken), 25 ng for the HCC827 cell line, and a range of dilutions of 15 ng, 10 ng, and 1 ng for the generated xenograft of the HCC827 cell line. Then, we added 1 µL of primers (Funakoshi Alu Seq, Bio-Rad, Cat No 10031261, Hercules, CA, USA) according to Funakoshi et al. [35]. The probe was labeled with fluorescent 6-carboxy-fluorescein (6-FAM). Ten microliters of TaqMan Universal Master Mix II were then added (Bio-rad, Cat No 1725281). The PCR conditions were 1 cycle of 95 °C for 10 min, followed by 50 cycles of 95 °C for 15 s, 56 °C for 30 s, and 72 °C for 30 s. The CT values were calculated using LightCycler^®^ 480 Software (Roche Life Science, Penzberg, Upper Bavaria, Germany) with the default settings. 

We performed the same qPCR on Alu sequences with distant organ samples (blood, CAM, and liver) from 3 chicken embryos. These organs were collected at the time of tumor resection. After collection (from veins in the CAM), blood was stored in a lysis buffer DL (Macherey-Nagel, France). Tissues (CAM and liver) were snap frozen and stored at −80 °C. Total DNA extraction was performed using the NucleoMag Tissue extraction kit (Macherey-Nagel, France). The protocol for qPCR of Alu sequences was the same as for tumors.

#### 2.9.3. NGS Analysis of Xenografts

Tumors obtained in ovo were also used for NGS analysis and comparison with patient biopsy FFPE samples. The protocol for library preparation and sequencing was the same as for cfDNA patient samples [34].

#### 2.9.4. Statistics

No statistical analysis was performed in this proof of concept.

## 3. Results

### 3.1. Description of the Patient Cohort 

The study’s design is available in the flowchart (Figure 1). We sampled 37 patients in our study, and 35 were enrolled. Two patients were excluded: one because of an experimental bacterial contamination, and another for major hyper leukocytosis, leading to unfeasible CTC enrichment. Lastly, six patients were included in both the breast and the prostate cancer cohorts, and 23 in the lung cancer cohort. 

The patient clinicopathological data included in our study are summarized in Table 1. 

Men and women represented 51% and 49% of the cohort, respectively. Median age was 66 years old. Current and former smokers represented a large majority in the lung cancer cohort (96%). This population of patients with metastatic (stage IV) disease was heavily pretreated overall, especially for breast and prostate cancer, with 4.5 and 3 median prior treatment lines, respectively. Forty percent of patients had more than three different metastatic sites, reflecting advanced diseases. Thirty-one percent had a significant impaired general condition with an Eastern Cooperative Oncology Group Performance Status (ECOG-PS) higher than 2. 

In the breast cancer group, the two patients with triple negative breast cancer were also carrying a BRCA genomic alteration. All the patients were negative for HER2 amplification or mutation. Unfortunately, Ki67 staining was only available on two of the six breast cancer patients. For both cases, Ki67 was high (25% and 90%) and in ovo engraftment was successful. 

In the prostate cancer group, median PSA blood level was 90 ng/mL, with a high degree of disparity between patients (Standard Deviation (SD ± 529). 

In the lung cancer group, adenocarcinoma was the most frequent histological subtype (48%). Fourteen were positive for PD-L1 in FFPE tissue at diagnosis, and the median percentage of positive tumor cells for PD-L1 staining in the original FFPE sample was around 5%. None harbored targetable molecular alterations.

### 3.2. Enrichment of Mimicking CTCs

Cell lines of epithelial origin were used to evaluate the enrichment efficacy of the ClearCell FX1 device. PC-3 (CRL-1435™, ATCC) is a cell line initiated from a bone metastasis of a grade IV prostatic adenocarcinoma from a 62-year-old male. A549 (CCL-185™, ATCC) were isolated from the lung tissue of a 58-year-old male with lung cancer. We used a small number (50–200 cells/7.5 mL of whole blood) of spiked mimicking CTCs to be representative of the rarity of the patient’s CTCs in the bloodstream. Only living fluorescent cells were counted. Overall, we obtained a recovery rate of 52% to 71% for these epithelial cell lines (Figure 2), in agreement with previously published work [14]. 

### 3.3. Immunostaining

As illustrated in Figure 3*,* we were able to identify and count CTCs, by a combination of different staining techniques (negative for immune biomarkers CD15, CD41, CD45 and positive for DAPI). In our hospital, for the histological exploration of lung cancers at diagnosis, the EpCAM biomarker is not integrated because there is a high variability in expression level, while PD-L1 expression is the routine predictive biomarker for anti-cancer immunotherapy and is always integrated in all lung tumor explorations. We performed a CTC count and PD-L1 expression assessment on a sample of the lung cancer cohort (Figure 2). Unfortunately, the procedure was not fully available at the beginning of the study, which prevents us from presenting the CTC count and PD-L1 evaluation on CTCs for all patients. Of the 16 patients in whom it was evaluated, we obtained a median CTC number of 57 cells per 7.5 mL of whole blood, with a wide heterogeneity (SD ± 87). Median percentage of CTCs with positive PD-L1 staining was 6% (SD ± 29%). Nine out of 16 patient samples (56%) were positive for PD-L1 staining (Figure 2). PD-L1 positivity in Formalin-Fixed Paraffin-Embedded (FFPE) tumor samples and in CTCs did not appear to be correlated. 

### 3.4. Engraftment Rate

The tissue culture step was performed in non-proliferative conditions. We enumerated the CTCs after enrichment, in a representative sample. The total count of cells (containing residual WBCs and CTCs) was performed before the tissue culture step, and it is not technically possible to enumerate CTCs before engraftment in ovo, in the absence of selection-positive markers, compatible with cell viability. We performed CTC-Derived Xenografts (CDXs) in 35 patients, with a total of 115 engrafted eggs. The mean number of engrafted eggs per patient was 3.3. At the time of the tumor resection, 71 eggs were alive and 44 were dead. The cause of death remained unknown in most of the cases. For five patients of our cohort, all eggs were dead at the time of analysis. We did not perform histopathological analysis in two cases, which were used for further explorations. Overall, 28 patients had at least one egg for histopathological examination and amongst these samples, 25 tumors were observed. More interestingly, we obtained a tumor from 20 different patients out of 33 (61%). Based on the population of patients with at least one living egg at the time of analysis, the engraftment success rate increases to 71% (20/28). There was no major difference between the three different tumor types, with an engraftment rate of 3/6 (50%) in the breast cancer cohort, 4/6 (67%) in the prostate cancer cohort, and 13/21 (62%) in the lung cancer cohort (Figure 4). 

According to the heatmap in Figure 4, we did not find any links between in ovo engraftment rate and patient characteristics such as tumor burden (number of metastatic sites), or number of previous treatment lines. Likewise, we did not find any links between the CTC count or culture time, and the engraftment success rate. Due to the small size and heterogeneity of our population, we did not perform a statistical analysis on these topics.

### 3.5. Anatomopathological Analysis and Immunochemistry

As expected, tumors obtained from HCC827 tumor cells were differentiated (glandular differentiation is seen, which is pathognomonic of adenocarcinoma) with positive TTF-1 staining. Both used the same protocol as for CTCs (Figure 5). 

Overall, 25 tumors were used for histopathological examination. CAM tumors did not retain the original morphological phenotype of the patient’s tumor. As illustrated in Figure 5, they were dedifferentiated (regardless of the tissue of origin), hypercellular lesional tissue proliferating entirely composed of spindle-shaped cells with fibrous or inflammatory background. The neoplasm cells contain eosinophilic cytoplasm with atypical oval nuclei. Nuclear pleomorphism was observed with vesicular nucleus, sometimes prominent nucleoli. Blood vessels can be observed lying near neoplasic cells. In most cases, the pathologist observed neither necrosis nor abundant vascularization. The PDX CTC tumor sizes were highly variable, with a median size of 1.5 mm (ranging from 0.3 mm to 5 mm) for the longest diameter of the tumor observed (Figure 4). We performed TTF-1 and CKAE1-AE3 immunostaining on six CAM tumor slides obtained from lung and breast cancers, selected according to the immunochemical profile of the original tumor. We obtained only negative stainings (Figure 5) in the breast and lung cancer cohorts. We therefore decided not to perform prostate-specific staining, as the tumor appeared as dedifferentiated as the two other tumor types. 

### 3.6. PCR Detection of Human DNA Alu Sequences 

We extracted DNA from two histologically proven CAM tumors, after annotation of an area with >70% tumor cells at pathological analysis, from distant organs of three embryos, and from a CAM tumor resulting in the engraftment of HCC827 tumor cells, used as a positive control for our experiment. Taqman probes were used for the quantitative PCR.

We obtained high-quality qPCR results with a linearity in the range of Cycle Threshold (CT) observed throughout the different DNA concentration inputs with the HCC827-derived tumors (Figure 6). Chicken DNA did not interfere with the reaction, as attested by both our linear results, and the weak signal obtained using only DNA from the chicken embryo. The positive result with the HCC827-derived tumor was expected, after we observed a differentiated tumor at histopathological examination. The CTs observed were proportional with the DNA concentration inputs: a mean CT of 22.3 (SD ± 0.2) for the 1 ng experimental condition, 18.8 (SD ± 0.2) for the 10 ng, and 17.1 (SD ± 0.2) for the 15 ng. As expected, we obtained the lowest CT, at 11.6 (SD ± 0.9), with the HCC827 cell line (without any engraftment procedure). Unfortunately, the amount of residual material for in ovo tumor derived from patient number 25’s CTCs was not sufficient to perform the qPCR. 

In the two tumor samples, from Patients 24 and 26, we obtained positive results with significant differences in CT between the chicken DNA negative control (chicken embryo without human cell engraftment) and in ovo CDX. Indeed, the mean CT differences (mean CT for the patient’s sample, subtracted from the mean CT for chickens) were 2.4 and 4.2, respectively, for Patients 24 and 26. These results show the presence of human DNA in the tumor we observed, at histopathological analysis. 

In the distant metastasis samples, the qPCR was also positive, which indicates the presence of human DNA in the embryo’s distant organs. We were able to identify the presence of human Alu sequences on the lower CAM, in the blood, and in the chicken embryo liver (Figure 6). We observed a heterogeneity in the distribution of Alu sequences between the different distant organs. The highest quantity of human tissue was found in the lower CAM in the three samples.

### 3.7. NGS Analysis

We conducted a feasibility test to explore the molecular profile of PDX CTC tumors, with the residual tissue materials from two independent PDX tumors, from Patient 13. 

We chose Patient 13′s samples for NGS analysis because the two in ovo xenografts were of sufficient size (in terms of extracted human DNA amount), with availability of the cfDNA sampled at the time of CTC enrichment, and availability of the FFPE original tumor of the patient. Using NGS testing, the TP53 c.215C > G constitutional genomic homozygous polymorphism was found in all the patient-derived samples analyzed. These included the FFPE original patient’s tumor, the cfDNA sampled at the time of inclusion, fresh CTCs sampled at the time of inclusion, and in ovo tumors 1 and 2 obtained after CTC engraftment (Figure 7).

In parallel, ctDNA and fresh CTC DNA sampled at the time of inclusion, as well as DNA derived from Patient 13′s in ovo tumor 1, contained a Kirsten Rat Sarcoma Viral Oncogene Homolog (KRAS) p.Gly12Val somatic mutation. This somatic alteration was not detected in Patient 13′s original FFPE tumor (sampled at the time of diagnosis) nor in Patient 13′s in ovo tumor 2 (Figure 7). In summary, we obtained a genomic concordance between the patient’s primary tumor, the liquid biopsy, and the resulting in ovo tumors in terms of constitutional polymorphism. Although found in one of the in ovo tumors, somatic KRAS alteration is found with greater heterogeneity (absent in the initial tumor sample, but present in liquid biopsies). 

## 4. Discussion 

Metastatic dissemination occurs via CTCs. This cell type displays a high diversity and heterogeneity. Some of these tumor cells are possibly implicated in the development of treatment resistance, making them of particular interest in the study of cancer resistance mechanisms and in drug development [13]. The availability of clinically relevant tumor models, as well as new pharmacological targets, is essential for improving the clinical success rate in drug development. The CAM assay for patient-derived xenografts in oncology is a valuable model, with significant advantages over other in vivo models such as low cost, rapid tumor growth, 3Rs compliance, reliability in reproducing the tumor microenvironment, and the possibility of a metastatic or angiogenesis study [3,4,6]. The in ovo model allows scientists to easily study the dissemination processes and infiltration of tumors via immune cells. In classical tissue culture conditions and for organoids, the nutritive medium needs to be continually renewed, which is only possible with a microfluidic system. It is well known that the in vitro culture conditions are associated with growth and response to drugs [36]. The CAM assay has already been used for various cancer types, but only one other team has performed in ovo CTC-derived xenografts [33]. We reported in this article the development of a method for enriching CTCs from blood samples, followed by engraftment onto the CAM, aiming towards the development of a drug testing platform. 

Both sensitivity and viability tests for the ClearCell FX1 system have been conducted in this project. A recovery rate of between 52% and 71% was observed with epithelial living mimicking CTCs (Figure 2). Our recovery rate was slightly lower than in previous studies, such as the 80% described by Lee et al. after ClearCell FX1 CTC enrichment [27]. We chose these experimental conditions (low number of mimicking cells spiked into 7.5 mL whole blood) to be representative of most clinical conditions. These findings suggest that the mesenchymal phenotype of the in ovo tumors cannot be attributed exclusively to the enrichment method.

Prior to the studies described in this article, a viability test was conducted on CTCs for a cohort of 12 NSCLC patients, using Trypan Blue Solution (TBS) staining. A viability rate of 96% (SD ± 4.9) was obtained (internal data).

Based on the presence of a tumor at histopathological analysis, we obtained an engraftment rate of 61% in ovo CDX (20/33) of the sampled patients. A high degree of heterogeneity was reported in previous studies using the CAM assay, with an engraftment rate ranging from 45% to 100% [9]. Pizon et al. performed in ovo xenografts with circulating Cancer Stem Cells (CSCs) from breast cancer and reported an engraftment rate of 50% in a small sample of 10 patients [33]. 

It is also relevant to point out that this is higher than the positive engraftment rate in the rodent model. The chicken embryo CAM model is also more time efficient, with 10 days between engraftment and xenograft removal as compared to up to 4 months in mice [3,4,6]. CDXs in mice were developed for different cancer types, such as Small Cell Lung Cancer (SCLC), NSCLC, melanoma, and prostate and breast cancer, with often low successful engraftment rates [37]. Although CDXs in mice offer added value for understanding tumor biology, the time required to generate xenografts, combined with the low success rate for engraftment, represents a significant drawback for this model. In ovo CDXs might be a complementary model in this field, with their shorter timeframe and higher engraftment rate [6]. 

We hypothesized that clinical features of tumor aggressiveness, such as impaired general condition, high tumor burden (≥3 metastatic sites), or heavily pretreated patients, could positively impact the engraftment rate. Indeed, an increased engraftment effectiveness was observed by Zhao et al. with highly aggressive pancreatic tumors, compared to less aggressive phenotypes [38]. No such association was identified in our cohort (Figure 4). 

In our study, the culture time did not seem to impact the engraftment rate (Figure 4). Pizon et al. showed an association between the absolute counts of CSC-derived tumorspheres and the engraftment success [33]. The median number of recovered CTCs per 7.5 mL of whole blood was 61 cells (SD ± 89). The variability was high, as had already been demonstrated for lung cancer in previous publications [14]. Here, the number of patients included in our cohort was too low to perform a statistical analysis. 

In 5 out of 33 cases (15%), all the engrafted embryos died before tumor resection. Aggressive disease features (impaired general condition, high tumor burden, patient heavily pretreated) did not appear to be associated with embryo death (Figure 4). This was therefore not in line with our hypothesis that aggressiveness of the tumor could have led to mortality through rapid diffusion of the disease in the embryo. The remaining immune cells engrafted with the CTCs could possibly have led to embryo death: a qPCR on DNA Alu sequences could allow us to better characterize extension of the disease to distant organs and the presence or not of human immune cells, in early dead embryos. On early dead embryos, we could also perform a transcriptomic analysis targeting immune-related signatures from a previously described immune-related gene set on distant organs [39]. In the literature, the reported embryo death rates are comparable to ours, ranging from 7% to 50% [6].

Surprisingly, no specificity related to the tissue of origin of the tumor was observed at histopathological analysis, although spindle cell proliferation was reported by our pathologists (Figure 5). Small tumors had a median length of 1.5 mm (SD ± 1.5 mm). We could not find any comparable results in the literature, since in ovo CDXs had not been performed until Pizon et al., in which the tumor size at histopathological examination was not reported [33]. It was unfortunately not possible to weigh them because of their small size, although this is a simple and reliable way to evaluate tumors in ovo [40]. We acknowledge that the tumor size evaluation method we used is very flawed, because it is strongly dependent on the location and orientation of the block section. Although these tumors do not match the original tumor phenotype, their spindle cell appearance does not support an immune origin of these cells. In Pizon et al., the CDXs were obtained from breast cancer patients, with a good concordance between the original histological phenotype of the tumor and the CDX [33]. They recruited early-stage breast cancers, whereas we only enrolled metastatic breast cancer patients, including two triple negative breast cancers. The cell culture time, of 14 days, was also significantly longer in their study under different culture conditions, including a medium enriched with growth factors, in a sphere-forming assay. The cells were characterized using typical combinations of markers for breast cancer stem cells (CD24−, CD44+, ALDH1+). These differences could help to explain the varying results we obtained in the histopathological examination of our CAM tumor xenograft, appearing to be dedifferentiated, as compared to the original tumor. 

In the immunochemistry analysis, we could not find any feature of tissue-specific differentiation, with no staining of TTF-1 and CKAE-1-AE3 on the CAM tumors derived from our lung cancer patients (Figure 5). Furthermore, TTF-1 staining was positive on the CAM tumor derived for the HCC827 lung cancer cell line, showing that there were no technical issues explaining the absence of staining in the cohort’s samples. In addition, well-differentiated tumors were generated from CTCs (notably from prostate and small cell lung cancer) in immunodeficient mice, and concordant immunostainings between primary tumors and in ovo tumors have already been obtained for different cancer types [6,30]. The mesenchymal state observed here might be linked with the CTCs’ biological nature, the CTCs’ specific isolation method, or the CTCs’ in vitro expansion before engraftment onto the CAM. Based on previous experiments conducted in our laboratory, the ClearCell FX1 device allowed better enrichment of live CTCs and successfully enriched epithelial cells (Figure 2). We therefore hypothesize that this mesenchymal state is explained by the Epithelial to Mesenchymal Transition (EMT) occurring in the metastatic process, and by the short timeframe. These factors could prevent reversion to the original phenotype at the time of microscopic examination and staining. Saito et al. suggested that TTF-1 might restore the epithelial characteristics of lung cancer cells. They showed a modified shape and increased E-cadherin expression after TTF-1 adenoviral transduction in what was originally a TTF-1-negative lung cancer cell line [41]. These findings are consistent with our results. Indeed, CTCs are quiescent in the blood and probably need time to undergo the Mesenchymal to Epithelial Transition (MET). This transition might also be more difficult in the chicken embryo microenvironment, particularly suitable for EMT, since genes involved in developmental EMT have also been shown to be implicated in tumor-related EMT as well as cell migration [42]. Overall, we hypothesized that the short embryo development time associated with the characteristics of the CAM microenvironment could have limited the ability of the tumor cells to proceed to MET, resulting in dedifferentiated tumors in the histopathological analysis and immunochemistry. Pizon et al. reported successful engraftment of breast cancer CSCs after a tumorsphere culture step, with a concordant immunostaining and histopathological aspect between primary tumors and in ovo tumors [33]. Even if it was performed on a small number of patients and with a different enrichment method, this additional 3D culture step might be crucial for obtaining differentiated in ovo tumors from CTCs. The most appropriate culture conditions for the development of these models have not been significantly developed, and few teams have succeeded in long-term culture with CTCs. Maintaining cells in suspension seems to be important, and an extra-cellular matrix (such as Matrigel^®^) is often added. The role of hypoxia also remains to be better defined [43]. Our model could also be refined by adding a 3D culture step after optimization of culture conditions, to increase the engraftment success rate, and provide sufficient time for the cultured cells to activate differentiation pathways. 

The small size of CTC primary xenograft tumors might be due to their dissemination to distant organs just after seeding. Indeed, the in ovo xenografts were composed of spindle cells, without the usual tissue-specific membrane differentiation markers. This mesenchymal phenotype is associated with a migration capacity during EMT. This dissemination of tumor cells in the chicken embryo has already been demonstrated by several teams. Most recently, Pawlikowska et al. showed the detection of fluorescent metastatic foci in the chicken embryo after engraftment of fluorescent cell lines [9]. To quantitatively explore this dissemination, we measured repetitive elements that are largely present in the human genome. Among them, Alu sequences are the most abundant, comprising around 10% of the human genome [44]. These sequences are human-specific and can be quantified via qPCR to detect human cells in immunocompromised mice after tumor transplant, and for a quantitative assessment of treatment efficacy [44,45]. This method is commonly used in the CAM assay to evaluate the metastatic dissemination of tumors after xenografting. Palmer et al. used qPCR amplification of human Alu sequences after DNA extraction from chicken embryo internal organs, to confirm the presence of metastasis in liver and lungs [46]. We performed a qPCR on Alu DNA sequences in the embryo’s distant organs for three patients in the lung cancer cohort, with evidence of human cells in the residual lower CAM, liver, and blood (Figure 6). These results support the idea of disease dissemination in the embryo, even if the presence of human DNA in distant organs in relation to DNA fragment diffusion rather than migrating cells cannot be excluded with this method. Further exploration could include transcriptomic analyses to define the characteristics of the cells that were able to metastasize in the embryo. 

Due to prior work to ensure the good recovery and viability rate of CTCs, the small size of the in ovo tumors probably did not come from these factors.

Moreover, using the same specifically designed probes, we also detected genomic Alu sequences via qPCR in two in ovo tumor samples, attesting to the presence of human tissue [35]. Unfortunately, we could not perform these Alu and NGS DNA analyses on all the CAM tumors, because of the small size of the available FFPE tissue. Regarding the very similar histopathological observations made in the xenografts across the different patients’ samples, we can therefore assume that these tumors come from the patients’ CTCs.

In the first instance, we explored DNA sequences through NGS analysis in one patient (Figure 7). We carried out this analysis on a different patient from those on which we had performed the Alu sequence quantification, to have as much tumor material as possible. The germline polymorphism found in both in ovo tumors and in the original patient’s FFPE tumor allows us to ascertain with confidence that the observed in ovo tumors were derived from the engrafted patient’s CTCs. The allele frequency of this polymorphism is 74% in the European (non-Finnish) population [47]. 

The KRAS p.Gly12Val somatic alteration was found in the fresh CTCs, in the cfDNA, and for in ovo tumor 1. This confirmed the feasibility of an NGS analysis for xenografted CTCs. It also underlined the tumor heterogeneity with the KRAS Wild Type (WT) profile of the original patient’s FFPE tumor, and for in ovo tumor 2. The original patient’s tumor sample was obtained one week after CTC enrichment. Therefore, the discrepancy between the in ovo tumor and the patient’s tumor suggests a broad spatial heterogeneity in this patient’s disease and supports the use of several xenografts for each patient. The presence of this mutation in the cfDNA excludes the hypothesis of a genotypic drift due to the engraftment in the CAM. In a large database study, 27% of metastatic lung adenocarcinoma carried KRAS mutations, and KRAS G12V accounted for 18% of KRAS mutated patients. Patients with KRAS mutations had significantly poorer outcomes [48]. For chemosensitivity tests, clonal heterogeneity could limit the relevance of the results.

The strengths of our study mainly lie in the innovative protocol, which allows us to consider the possibility of developing an avatar from the patient’s tumor from a simple blood test. Our CTC enrichment method showed good performance, with a high recovery rate and viability rate after ClearCell FX1 runs. The engraftment results with tumor cell lines were consistent with the literature and allowed us to expect a satisfactory engraftment rate. We observed a mesenchymal aspect in the CAM tumors after CTC engraftment, as well as negative tissue-specific stainings. Here again, the results we obtained with the cell line-derived xenograft indicated that these features reflected the specific behavior of CTCs after engraftment, but not a technical issue. We obtained a definitive proof of the human origin of the in ovo CAM tumors, in a small sample of our cohort, with positive qPCR targeting human-specific Alu DNA sequences. This positive result was found in the CAM tumor but also in distant organs, supporting our hypothesis of dedifferentiated cells with a mesenchymal profile and a likely preserved migration ability. Finally, we demonstrated the feasibility of NGS analysis in this assay, with genomic concordance between the in ovo tumor and the original patient’s tumor, for constitutional polymorphism and somatic alteration. However, our model also has limitations to be addressed. First, the possibilities for analysis are limited due to the small size of the in ovo tumors and the small number of in ovo tumors per patient. It will therefore be necessary to repeat these analyses on a larger number of samples in the future. Secondly, comparisons with the patient’s tumor are limited due to the chicken embryo microenvironment, which might have led to a dedifferentiated phenotype of the tumor cells. In response to these limitations, the development of a 3D organoid culture step, with a longer culture time and optimized culture medium, could allow us to engraft a larger number of CTCs in each egg, with more time left for the CTCs’ Mesenchymal to Epithelial (MET) transition. We could then repeat these experiments to confirm the results presented here in a larger number of xenografts.

## 5. Conclusions

To our knowledge, we have performed the first xenografts from CTCs onto embryonated chicken eggs’ chorioallantoic membrane, with evidence of CTC dissemination to distant organs. DNA sequence exploration of the CAM tumor through NGS showed a genomic concordance with the original patient’s tumor and its liquid biopsy. Our engraftment rate was satisfying, but tumors were of very small size. No clear association was found between the original tumor features, or the patient’s characteristics, with the engraftment rate. We observed major differences in histopathological patterns when compared to original tumors. We hypothesized that these differences might be related to the short experimental timeframe and the chicken embryo microenvironment, possibly enhancing the mesenchymal characteristics of the CTCs. We plan to implement a longer 3D culture step before engraftment. These improvements could allow us to improve engraftment outcomes and start drug testing using this platform, with a direct comparison to the patient’s outcome. We also aim to use this innovative platform to study resistance mechanisms to anticancer drugs and metastatic dissemination pathways through transcriptomic analysis in CDXs.

## 6. Patents

Partial results reported in this manuscript were used for the application of patent “WO2020/089560A1; 7 May 2020”.

## Figures and Tables

**Figure 1 cancers-14-04085-f001:**
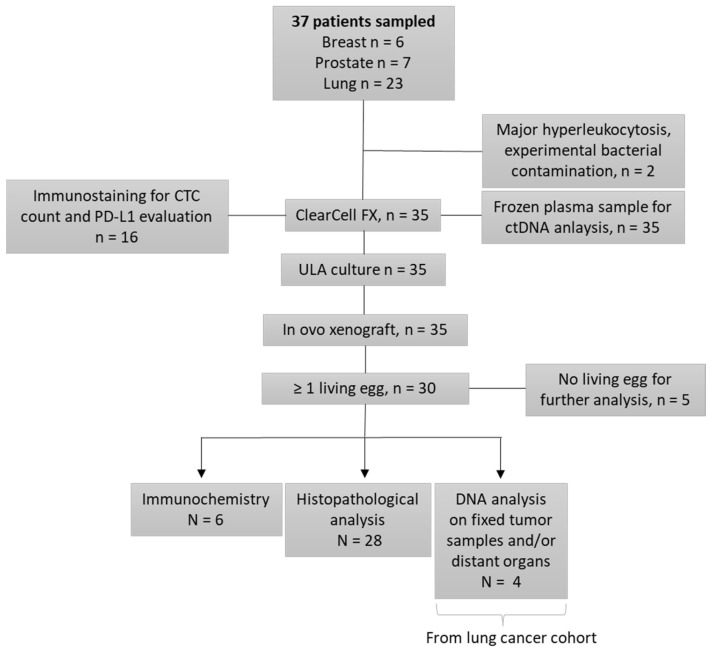
Study Flowchart.

**Figure 2 cancers-14-04085-f002:**
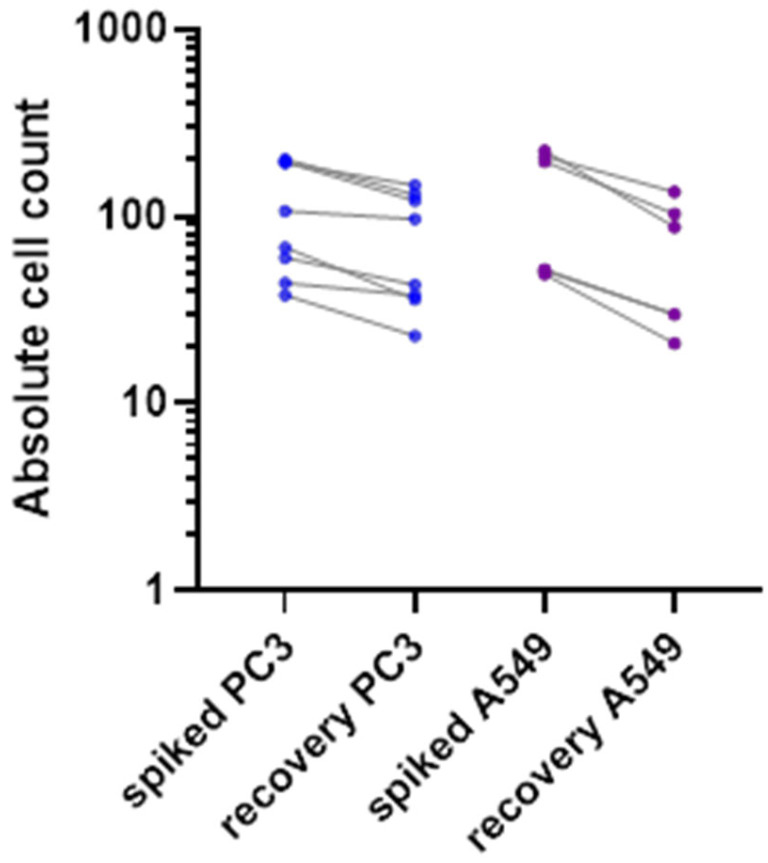
Recovery rate of two spiked fluorescent epithelial tumor cell types (PC3 in blue and A549 in purple), in healthy donor blood, with Clearcell FX1 device enrichment.

**Figure 3 cancers-14-04085-f003:**
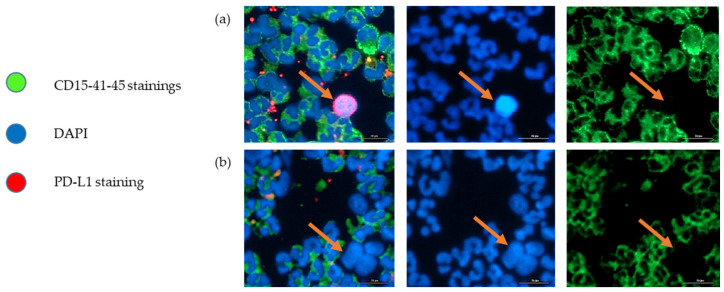
Representative images of immune cells (CD15+, CD41+, CD45+, DAPI+), and CTCs (CD15-, CD41-, CD45-, DAPI+, PD-L1+/−) of the sample from patient 28, after ClearCell FX1 enrichment. The arrows indicate a CTC. (**a**) PD-L1-positive CTC, (**b**) PD-L1-negative CTC. Scale bar = 20 µm.

**Figure 4 cancers-14-04085-f004:**
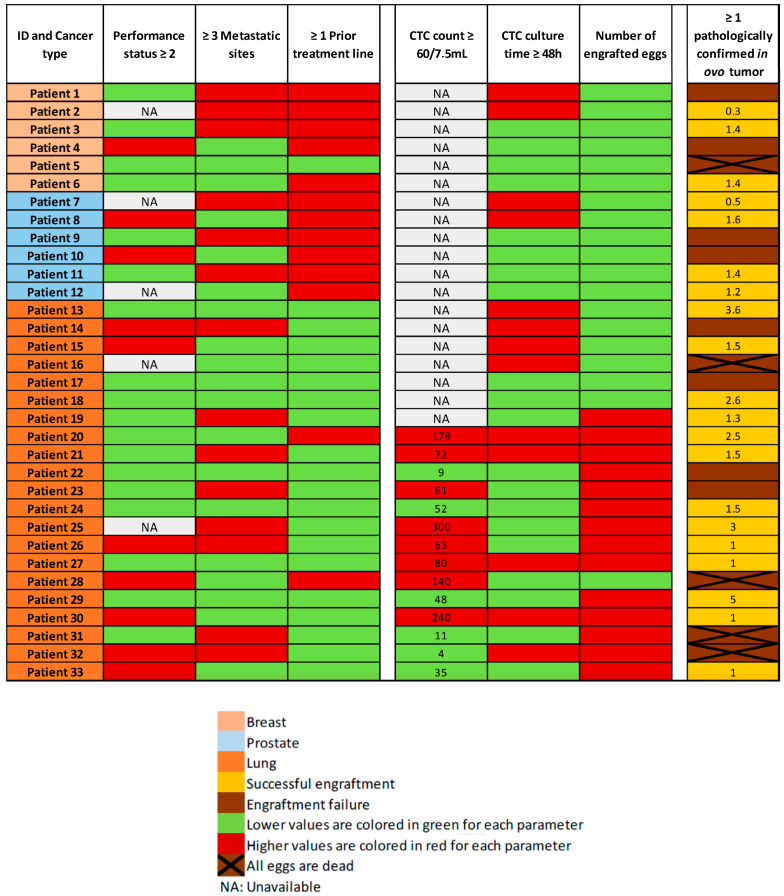
Heatmap of the main clinical characteristics of the patients, CTC counts (with the number of CTCs/7.5 mL of whole blood when available), CTC culture time, and engraftment outcomes (with the tumor diameter in millimeters observed at histopathological analysis, when available).

**Figure 5 cancers-14-04085-f005:**
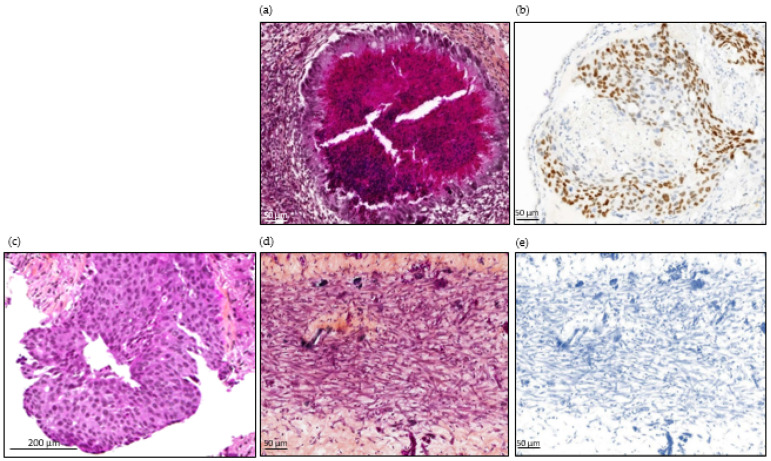
Histopathological images of tumor from patient number 24, at diagnosis and in ovo tumors from cell line and CTCs. (**a**) H&E staining of an HCC827-derived in ovo tumor. (**b**) TTF-1 immunostaining of an HCC827-derived in ovo tumor. Scale bar = 50µm (10× magnification). (**c**) H&E staining of lung biopsy from patient 24: invasive tumor composed of sheets of tumor cells that lack acini, tubules, and papillae with mucin production. Focally, cribriform patterns can be seen. Scale bar = 200 µm (10× magnification). (**d**) H&E staining of in ovo tumor derived from patient 24’s CTCs. (**e**) TTF-1 immunostaining of in ovo tumor derived from patient number 24’s CTCs. Scale bar = 50 µm (10× magnification).

**Figure 6 cancers-14-04085-f006:**
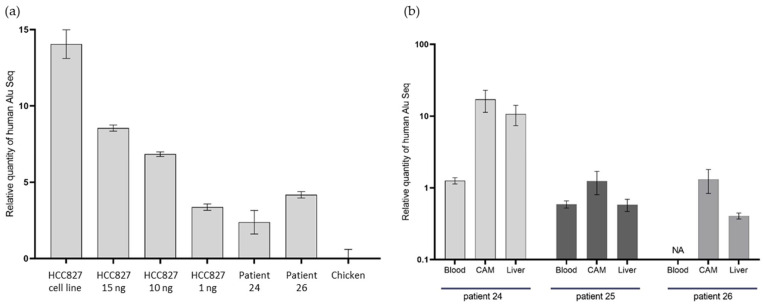
qPCR of DNA Alu sequences in in ovo tumors. (**a**) qPCR on DNA Alu sequences in tumors from Patients 24 and 26, with a DNA concentration range from HCC827-derived xenografts, using the cell line as a positive control and chicken tissue as a negative control. (**b**) qPCR on DNA Alu sequences in chicken embryo distant organs, following CTC engraftment from Patients 24, 25, and 26. NA: unavailable.

**Figure 7 cancers-14-04085-f007:**
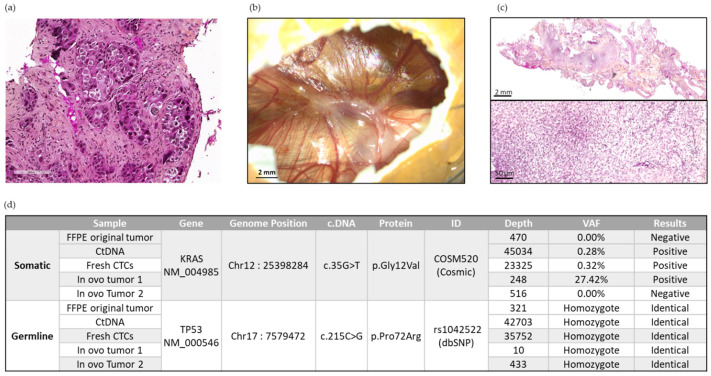
Representative panel of analyses carried out on Patient 13. (**a**) Representative image of the original tumor of Patient 13 at diagnosis. Scale bar = 200 µm. (**b**) Representative image of in ovo CTC-derived xenografts from Patient 13 at EDD18. (**c**) The 2× and 20× magnification of H&E staining of an in ovo CTC-derived tumor from Patient 13. Scale bars = 2 mm and 50 µm. (**d**) DNA sequencing exploration using the NGS method on different samples of Patient 13.

**Table 1 cancers-14-04085-t001:** Clinicopathological characteristics of the study population.

	Breast n = 6	Prostate n = 6	Lung n = 23	Total n = 35
Women—n (%)	6 (100)	0 (0)	11 (48)	17 (49)
Men—n (%)	0 (0)	6 (100)	12 (52)	18 (51)
Age—mean	56	72	68	66
≥5% weight loss—n (%)	1 (17)	4 (67)	14 (61)	19 (54)
Ever smoker—n (%)	0 (0)	2 (33)	22 (96)	24 (69)
ECOG-PS ≥ 2—n (%)	1 (17)	2 (33)	8 (35)	11 (31)
Stage IV	6 (100)	6 (100)	23 (100)	35 (100)
≥1 prior treatment lines	5 (83)	6 (100)	3 (13)	14 (40)
≥3 metastatic sites	3 (50)	3 (50)	8 (35)	14 (40)
Positive hormonal receptors—n (%)	4 (67)			
HER2 positive—n (%)	0 (0)			
Triple negative breast cancer—n (%)	2 (33)			
BRCA mutated—n (%)	2 (33)			
PSA (ng/mL)—median		90		
Gleason score at diagnosis—median		7		
Lung adenocarcinoma—n (%)			11 (48)	
Lung squamous cell carcinoma—n (%)			8 (35)	
NSCLC—n (%)			4 (17)	
Median % of PD-L1+ tumor cells in the original FFPE tumor sample			5	
Number of patients with positive PD-L1 staining in the original FFPE tumor sample—n (%)			14 (61)	
Targetable genomic aberration—n (%)			0 (0)	
CTC count/7.5 mL of blood—median [min; max]			57 [4; 300]	
Median % of PD-L1 + CTCs			6	
Number of patients with PD-L1+ CTCs—n (%)			9 (56)	

## Data Availability

The data presented in this study are available on request from the corresponding author.

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
