# Peer review of "Embryonated Chicken Tumor Xenografts Derived from Circulating Tumor Cells as a Relevant Model to Study Metastatic Dissemination: A Proof of Concept"

_cancers, 2022, doi:10.3390/cancers14174085_

Round 1

Reviewer 1 Report

The manuscript titled “Embryonated Chicken Tumor Xenografts Derived from Circulating Tumor Cells as a Relevant Model to Study Metastatic Dissemination: a Proof of Concept” describes the authors investigated patients’ circulating tumor cells using in ovo model. The qualities of the figures need to be improved. For the chicken embryo tumor model, I’d suggest the authors state that this technology has been used for decades. The followings are some concerns and comments have been pointed out that the authors may want to consider.

1) Line 159: I’d suggest the authors use “years old” for age instead of “years”.

2) Line 258: Did authors detect hypoxia condition after in ovo xenograft? This is important for the CAM model.

3) Lines 265-268, the authors stated CTCs grafted onto the CAM on EDD9; line 272, tumors began to be detectable on EDD10. Please confirm. Please note lines 812-813, the authors stated: “tumors were of very small size”.

4) Lines 273-274: the tumors were harvested at EDD18. Why did the authors choose EDD18? Did the authors consider EDD19-20 just one or two days ahead of hatching? Please provide tumor volume.

5) Line 437 Figure 2: a) Please add scale bars. b) Please include a color legend description in the figure legend.

6) Line 524 Figure 4: a) Please add scale bars. b) Please add a more detailed description in the figure legend.

7) Line 608 Figure 6: a) Please add scale bars. b) Please specify how many days. c) Did the authors even detect the red blood cells inside of the blood vessels on the tumor? This is important to prove functional blood vessels. d) Figure 6C table: Please use “0.28%” instead of “0,28%”. Please Check throughout the manuscript.

8) Line 865: Where are the references after [34]? All the [35] to [46] references are missing.

9) Please confirm there is no statistics analysis in the manuscript. Please provide if the authors have the related results.

10) Please confirm that DNA sequencing results only include one patient. Please provide if there are some others.

Author Response

Dear reviewer,

First we would like to thank you for the interesting and pertinent sremarks and questions about our work.

Please find in attached Word documents our answers.

Best regards,

Léa Payen et al.

Reviewer 2 Report

1.     This article focused on CAM xenograft derived from CTCs as a relevant model to study metastasis dissemination. However, the data was too fragmented and the manuscript was lengthy.  

2.     Of the collected patients, only the successful rates of CAM xenograft were relatively complete. The other data were just examples of patients such as DAN sequence or NGS analysis. Only three lung cancer patients presented distant metastasis to liver from CAM xenograft. Others were unknown. The data of patients with breast or prostate cancer were not related to this article. Too many missing data caused the lack of evidences to support the application of CAM xenograft derived from CTCs in metastasis dissemination study.

3.     If the research focuses on lung cancer to obtain complete data including CTCs, CAM tumor and distant organ metastasis from every collected patient, it would make more sense.

Author Response

(The authors gave the same response as above.)

Reviewer 3 Report

The authors report in this article the development of a way to purify CTCs from blood followed by an engraftment into the CAM of chicken embryo, with the ambition to go towards the development of a drug testing platform. Although interesting, the author’s data are far from this objective, and in fact, the way they perform CTCs isolation maybe compromising the way they will reach their goals.

Also, this topic is not completely new since recently Pizzon and collaborators have done exactly the same thing, with the exception on the way CTCs were isolated and expanded in vitro before implementation in the CAM. The only novelty is related with the identification of metastasis using this in ovo CTC-derived xenografts, which is a very interesting feature but very few cases were tested.

Thus, it it is my opinion that for the paper to be accepted needs a profound revision and some extra experiments to really prove the authors point on the development of a new protocol using freshly isolated CTCs from patients with metastatic cancer, which could provide a reliable tumor model after a CAM xenograft.

Major points to be addressed:

1.       During the 2-3 days of culture (before engrafting into the CAM model) do you observe any CTC increased cell number? Why did you not tested the previously described conditions for CTCs enrichment as a way to comparison?

2.       Why did the authors just used anti-PD-L1 for CTC staining? Did you tested for other CTC markers, such as EPCAM?

Do you have any explanation for the lack of correlation between PD-L1 on FFPE and CTCs? Could it be that the CTCs isolation protocol is being selecting specific CTCs phenotype?

3.       I think this will show that the isolated CTCs through this microfluidic device is enriching for isolated and mesenchymal CTCs and excluding Epithelial CTCs, and therefore, when these cells are engrafted in the CAM, they form mesenchymal tumors, not representative of the primary one. In fact, the CTCs isolation method seems to me as one of the main drawbacks for the proposed protocol.

4.       Refer the reason why did you use TFF-1 and AE1/AE3 to characterize the xenografted tumors. TFF-1 is not a differentiation marker in breast can in prostate cancer.

5.       In Page 12, you refer that the CAM tumors did not retain the original morphological phenotype of the patient´s tumor. Please show an image, and add a table with the histological comparison between patient tumor and CTCs CAM tumor.

6.       The authors should add a table with a description of all histopathological analysis done by the pathologist for each generated CAM tumors, as well as the respective primary tumor.

7.       The 3.3. Section is poorly described. It gives just a general description, but a more detailed one is required.

8.       The results from section 3.5 are poorly described. Once again, no clear comparison between the primary tumor vs CAM is shown nor discussed. Please clarify this information.

9.       Does the metastatic dissemination observed correspond from what is seen in the patients? Why did you not recovered for chicken lungs?

10.   In the conclusion the authors state that the DNA sequence exploration of the CAM tumour through NGS showed a genomic concordance with the original patient’s tumour and its liquid biopsy. This information is not clearly stated and described in the results section.

11.   The authors observed major differences in histopathological pattern when compared to original tumours. They hypothesized that these differences might be related to the short experimental timeframe and the chicken embryo microenvironment, possibly enhancing the mesenchymal characteristics of the CTCs. Nevertheless, I think that the explanation goes far beyond that, and must be indeed related with the biological nature of the recovered CTCs. Also the authors state that they plan to implement a longer 3D culture step before engraftment, and in fact they should at least show one or two cultures to see if they improve the protocol.

12.   Importantly enough, I think it would be crucial to compare other methods for CTCs isolation, in order to really understand if this mesenchymal state has to do with CTC biological nature, CTC specific isolation method, CTCs expansion in vitro before implementation in the CAM. All this aspects should be better

Minor points:

1.       Please indicate the number of CTCs engrafted in the CAM model

2.       In Figure 2, you show an example an atypical cells, but you should also add an image where you identify a CTC (or put a diffently colored arrow to indicate a CTC).

3.       In Figure 5 please add a title. Please indicate what NA means.

4.       Add a title to Figure 4 and 6

Author Response

Dear reviewer,

First we would like to thank you for the interesting and pertinent remarks and questions about our work.

Please find in attached Word document our answers.

Best regards,

Léa Payen et al.

Reviewer 4 Report

General comment: this is an interesting report that decipher a new model of ex vivo human tumor that have potential implication to study drug resistance. Indeed, this model should be improved as the observed characteristics of xenograft differs from original tumors. 

Other comments:

Abstract:

1)      please define “PDX”

Introduction:

1)      The potential superiority of the present model to a model of in vitro spheroid culture of CTC should be better explained. Why the culture in CAM is superior to in vitro medium?

Methods:

1)      Characterization of CTC was define by negative staining for immune cell markers and positive DAPI, but why performed a staining for PD-L1?

2)      The observed xenograft are small, is there any observation beyond D18 with larger tumor?

3)      Could it be exclude that presence of human DNA in distant organ may be related to DNA fragment diffusion rather than migrating cell?

4)      Presence in distant organ of human DNA did not preclude of the viability of tumor cell in distant organ. Is there any observation beyond D18 that support the development of distant metastasis

5)      Small number of analysis of distant DNA detection and NGS analysis is a concern

Results:

1)      The rate of patients with CTC is not clearly given

2)      The results of PD-L1 expression is not given in figure 3

3)      Results of the patient’s 25 DNA detection should be given in the revised version

4)      The discrepancy between the KRAS mutation observed in the two in ovo tumor underline the clonal heterogeneity of tumors and the need of several xenograft for the same patient

Discussion:

1)      The interest of the CAM model to determine different drug sensitivity for personalized treatment is easy to understand. In case of clonal heterogeneity this would be more challenging.

2)      Moreover, it could be more develop in the discussion for which type of drugs this model will be useful. Indeed, it would be interesting to perform a chimiogram for cytotoxic drugs, but other in vitro model are already used. For targeted therapy, the mutation analysis on ctDNA is already very informative.  Finally for immunotherapy the model is lacking immune cell and the mesenchymal phenotype of tumor cell is a concern.

3)      I don’t see the advantage of the CAM model for dissemination study? It will not drive the individual treatment as prevent dissemination is not an endpoint for patient care. The endpoint is to cure dissemination with efficient drugs. For mechanism comprehension of dissemination the CAM model seems less physiological than animal model.   

Author Response

(The authors gave the same response as above.)

Round 2

Reviewer 1 Report

Thanks for the response and the revised manuscript.  The review does not have further comments. Good luck. 

Reviewer 3 Report

The authors answered to all the points raised by the reviewer.

Reviewer 4 Report

No more comment